# Iodine(III)-Containing Reagents in Photo-Assisted and Photo-Catalyzed Organic Synthesis

**DOI:** 10.3390/molecules30040784

**Published:** 2025-02-08

**Authors:** Jaime G. Ibarra-Gutiérrez, Luis A. Segura-Quezada, Edson D. Hernández-Velázquez, Ana K. García-Dueñas, José A. Millán-Cortés, Kevin Mondragón-Hernández, Luz K. Miranda-Navarrete, Evelyn M. Valtierra-Camarena, Steffi Y. Yebra-Rivera, Omar E. Alférez-Carmona, Oliver E. Ávalos-Otero, Rubén Chávez-Rivera, Claudia de León-Solís, Rafael Ortíz-Alvarado, César R. Solorio-Alvarado

**Affiliations:** 1Departamento de Química, División de Ciencias Naturales y Exactas, Universidad de Guanajuato, Campus Guanajuato, Noria Alta S/N, Guanajuato 36050, Gto., Mexico; jg.ibarragutierrez@ugto.mx (J.G.I.-G.); la.seguraquezada@ugto.mx (L.A.S.-Q.); ed.hernandezvelazquez@ugto.mx (E.D.H.-V.); ja.millancortes@ugto.mx (J.A.M.-C.); 1436204j@umich.mx (K.M.-H.); lk.mirandanavarrete@ugto.mx (L.K.M.-N.); em.valtierracamarena@ugto.mx (E.M.V.-C.); sy.yebrarivera@ugto.mx (S.Y.Y.-R.); oe.alferezcarmona@ugto.mx (O.E.A.-C.); oe.avalosotero@ugto.mx (O.E.Á.-O.); 2Instituto de Ciencias Químico Biológicas, Universidad Michoacana de San Nicolás de Hidalgo, Av. Universidad S/N, Morelia 58000, Mich., Mexico; ana.garcia@umich.mx; 3Facultad de Químico Farmacobiología, Universidad Michoacana de San Nicolás de Hidalgo, Tzintzuntzan 173, col. Matamoros, Morelia 58240, Mich., Mexico; ruben.chavez@umich.mx; 4Escuela de Estudios de Postgrado, Facultad de Ingeniería, Universidad e San Carlos de Guatemala, Guatemala 01012, Guatemala

**Keywords:** photochemistry, iodine(III)-based reagents, photoredox, photocatalysis

## Abstract

Iodine(III) reagents have become a highly relevant tool in organic synthesis due to their great versatility as strong but green oxidants. Several transformations involving cyclizations as well as functionalization of different organic cores have been broadly described and reviewed. Herein, the participation of these reagents in photochemical transformations exclusively by direct irradition or in photoredox cycles using some transition metals, will be briefly described as well as some plausible further transformations that potentially can be developed.

## 1. Introduction

Iodine(III) reagents have been an excellent tool in organic synthesis. They have found exceptional applications as non-transition-metal-based oxidants, which have been broadly reviewed [1,2,3,4,5,6]. Their low toxicity and green (poor environmental-contaminant) oxidative features makes them good candidates for reactions such as halogenation [7,8,9,10], nitration [11], arylation [12], and amination which are among some of the most representative.

These transformations are either polar or pericyclic reactions which, in general, require heat or a metal for their activation. Herein, we will summarize in a concise manner, the most representative reactions involving the use of iodine(III) reagents in photochemistry from 1984 up to now. Thus, this overview covers the activation and functionalization of different bonds by direct photoexcitation as well as photocatalysis using the most reported transition metals. Throughout this review, several useful transformations, with particular emphasis on those involving the activation of strong C-H bonds present in different organic functionalities, will be discussed to put into perspective the relevance of this synthetic strategy in organic chemistry nowadays. Other photochemical reactions using iodine(V) reagents were out of the scope of this review [13].

The aim of this document is to contextualize the great advances of this synergic combination of iodine(III) reagents and photochemistry and give a plausible perspective for the future on this type of chemistry.

## 2. General Considerations

Strategies in organic synthesis to access different structural motifs and relevant frameworks present in all living species, are broad and diversified [14,15,16,17,18,19,20,21,22,23,24]. Among some of the most important transformations to consider for synthetic methodology development, it is possible to highlight the activation and sequential functionalization of inert and stable C-H bonds [25,26], which is a straightforward strategy to introduce different groups in a molecule. Organometallic approaches using transition metals are some of the most common in this regard. However, the process typically requires strong reaction conditions such as high temperatures or high catalytic charges to complete the cycle in these metal-catalyzed transformations. Depending on the groups present on the molecule, this issue could represent an inconvenience to consider for the overall strategy. The challenge regarding the direct C-H activation could be envisioned considering the following bond energies [27] (Figure 1).

Few synthetic tools can provide energy levels high enough to carry out the previous and direct functionalizations. One of the most relevant protocols involves the use of the innate energy from light. In this context, the energy in kcal·mol^−1^ supplied by a specific wavelength can be calculated according to the fundamental equation (Equation (1)):(1)$$E\;=\;h\;\tilde{v}\;=\;2.86\;\times \;10^{4}/\lambda \;(\mathrm{in}\;{\mathrm{cm}}^{-1})$$

Thus, it is easy to calculate that, light of *λ* = 210 nm gives 136.2 kcal/mol, which is enough energy to break even the strongest _sp_C-H bond [28]. In consequence, light in chemistry has been broadly combined with other synthetic tools. Herein, we aim to present a concise review about the synthetic applications of photochemistry in combination with iodine(III) reagents to provide a better understanding of direct photoexcitation and photocatalysis.

## 3. Synthetic Methods Using Direct Photoexcitation and Iodine(III) Reagents

As stated previously, the most representative synthetic methods using iodine(III) regents involved in direct photoexcitation will be reviewed first.

The earliest report of the combination of iodine(III) species with photochemistry was described in 1984 by Suarez’s group [29]. The procedure involved irradiating diacetoxyiodobenzene (PIDA) in the presence of molecular iodine under white light with a W-lamp for the homolysis of alcohols **1** to form new C-O bonds via [1,5]-H shift. This was applied originally to get cholesterol derivatives **2**. The reaction nowadays is known as the Suarez-cleavage. This procedure also allowed the formation of carbonyl derivatives from cyclic carbohydrates **3** to get ring opening products **4** [30]. On the other hand, due to the great method’s versatility, the procedure has been broadly exploited and applied to the synthesis of cyclic ethers [31] as well as for chiral spiroacetals from pyranose or furanose via alkoxy radical intermediates [32]. Several other uses have been described with this radical fragmentation. Concerning the mechanism to form C-O bonds in steroids, this started with PIDA homolysis under light irradiation, forming iodanyl **5** and the acetoxyl radical. The former reacted with molecular iodine to form acetyl hypoiodite along with an iodine radical that reacted with sugar **3**, generating oxygen-centered radical **6**. This formed radical formate **7** through hexose ring opening. The final reaction with hydriodic acid yielded the observed product **4** with regeneration of the iodine radical that continues the catalytic cycle (Figure 1).

The next year, Varvoglis and co-workers described the UV-photoexcitation of zwitterionic iodonium salts **8** which gave 1,3-oxathiole-2-thione **9** in reaction with carbon disulfide [33]. Even though the yields were low, irradiation dramatically decreased the reaction time from weeks to hours, indicating a different activation of the iodonium yilde (Equation (2)).
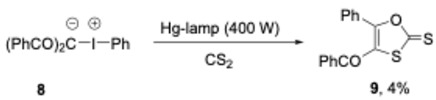
(2)

That same year, the group of Varvoglis [34] also described the photoexcitation of PIDA and [bis(trifluoroacetoxy)iodo]benzene PIFA under the 400 W Hg-lamp irradiation that were used as initiators in the *i*-butyl (vinyl ether) polymerization (Equation (3)).
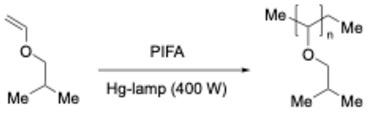
(3)

Later, Minisci and co-workers [35] described alkyl radical generation via decarboxylation of carboxylic acids **11** that reacted with pyridinium salts **10**, using a low-pressure Hg-lamp to get C2 functionalized products **12**. Herein, direct irradiation on PIDA derivatives **13** gave rise to iodine-centered radicals **14** that, after release of carbon dioxide, easily homolyzed to give alkyl radicals **15**. These reacted with pyridine derivatives **10** to give C2-functionalized carbon-centered radical **16** which was oxidized by **14**. During this process, aromaticity was recovered, yielding the observed final pyridines in their salt form (Figure 2).

Then, in 1995, Togo and co-workers [36] described the synthesis of adamantyl sulfides **19** by direct irradiation of [bis(1-adamantanecarboxy)iodo]arenes **17** with a high-pressure Hg-lamp in the presence of disulfides **18**. The developed procedure took place via radical decarboxylation of the adamantyl-containing iodine(III) reagent **17** to give a carbon-centered radical **20**, which reacted with disulfide 18 to finally give observed sulfides **19a**–**d**. The protocol tolerated a good variety of functional groups on the aryl moiety and proceeded in mild reaction conditions (Figure 3).

Two years later, the same authors reported [37], another photochemical homolytic cleavage on iodine(III) reagents. In this development some o-alkyl and o-aryl-benzenecarboxylic acids **21** and alcohols were subjected to Suarez’s conditions to yield heterocycles such as **22**. Direct irradiation of hypoiodite **24** formed in situ from iodine(III) derivatives **23**, led to the formation of several oxygen-centered radicals **25** that reacted intramolecularly via [1,5]-HAT in a direct *_sp_*^3^C-H bond activation of the o-alkyl group to form the corresponding benzylic carbon-centered radical **26**. Then, the reaction with molecular iodine gave **27**. The final intramolecular cyclization gave rise to the formation of different cumarine, phtalide, and benzocumarine derivatives **22a**–**i** with concomitant hydroiodic acid release. An excellent scope was presented in this work (Figure 4).

In 2000, Hadjiarapoglou and co-workers [38] carried out the direct photoexcitation of phenyl iodonium ylide **28**, obtained from dimedone using a 400 W medium pressure Hg-lamp, in the presence of different alkenes **29**. The reaction led to the formation of dihydrofuranes **30** in good to high yields via cycloaddition. This process was regio- and stereoselective, highlighting the isolation of a single isomer where the oxygen of the ylide bonded exclusively at the more substituted side of the alkene in the final products **30a**–**e** (Figure 5). Interestingly, the olefin stereochemistry was not preserved, which implied a stepwise reaction mechanism.

In 2009, Matveeva and co-workers [39,40] carried out the synthesis of several λ^5^-phoshinolines, **34** and **35**, by photochemical reaction between acetylenes **33** and mixed phosphonium-iodonium ylides **31** or **32**. This rare class of heterocycles was obtained under direct irradiation with a Hg-lamp at 366 nm. The few protocols available at the time to obtain aromatic phosphorus derivatives **34a**–**f** in very mild reaction conditions were key features. Incorporation of a thiophene ring in the mixed ylide **31** was a variant of the method that enabled the formation of corresponding thiophene-fused λ^5^-phoshinolines **35a**–**c**. The full mechanistic investigation regarding this photochemical reactivity was reported the following year [41,42] (Figure 6).

Direct hydroacylation using branched aldehydes as starting materials via acyl radical formation was a tremendous challenge due to the easy decarbonylation of formyl substrates. Accordingly, a practical solution to this problem was described by Maruoka’s group in 2014 [43]. In this work, different acyl radicals **42** were softly generated from the reaction of branched aldehydes **36** with iodanyl **41** and/or acyloxy radicals **40** which were obtained from photochemical homolysis of iodine(III) derivatives **38** under visible light irradiation (400 nm). Then, protonated iodine(III) derivative **43** was formed along transient acyl radical **42** which was trapped by dimethyl maleate derivatives **37**. The former reaction led to the formation of the corresponding adduct with radical transfer to the maleate moiety to get carbon-centered radical **44**. Finally, in a radical pathway, **43** carried out a HAT regenerating iodanyl radical **41** which continued in another cycle with the concomitant formation of observed products **39a**–**f** (Figure 7).

The following work presented by Li and co-workers [44], described the synthesis of ynones **47** using 2-oxo-2-phenylacetic acid derivatives **45** and terminal bromoacetylenes **46** in the presence of catalytic amounts (30 mol%) of hydroxybenziodoxolone (BI-OH). This iodine(III) reagent was photoexcited using sunlight or a blue LED (455 nm) at room temperature. The reaction mechanism started with the interaction between BI-OH and **45** to bond the oxalic functionality to iodine(III) reagent obtaining **48**. This intermediate homolyzed giving iodanyl radical **49** and oxygen-centered radical **50**. In one side, **49** reacted with bromoacetylenes **46** forming new iodine(III) reagent **52**. On the other hand, homolytically fragmentation of **50** released carbon dioxide and acyl radical **51** which reacted with **52**, leading to the formation of iodine(III) adduct **53**. Next, a radical cleavage released observed products **47a**–**h** with the concomitant formation of **54** after the reaction with radical bromine. Final hydrolysis regenerated BI-OH. In general, this protocol represented a novel approach in organic synthesis in terms of the energetic economy considering the endless sunlight. The broad scope of this procedure and good chemical yields were the most representative features (Figure 8).

In 2016, Wang and co-workers [45] described the decarboxylative acylarylation of acrylamides **55** using α-oxocarboxylic acids **56** in the presence of catalytic amounts of BI-OH as the iodine(III) reagent. This procedure was applied to the synthesis of several 2-oxoindolines **57** with different α-acyl aromatic derivatives. The optimal conditions implied the irradiation of BI-OH with a blue LED obtaining good to excellent yields. Limitations included the presence of amino and some ester functionalities on the aromatic ring. In general, the reaction mechanism involved the formation of acyl radical **59** via blue LED irradiation promoted decarboxylation of iodine(III) oxalic derivatives **58** formed by reaction of BI-OAc with **56**. On one side, acyl radical **59** was formed which reacted with **55** in a Markovnikov fashion, leading to the acyl group bonded to the double bond with the concomitant indoline formation. Ring closing ensued with the formation of non-aromatic aryl radical 60. On the other side, after decarboxylation, iodanyl radical **49** was also formed which aromatized the previously formed radical to continue with another cycle along the formation of observed products **57a**–**e** (Figure 9).

In 2018, Muñiz’s group [46] described an interesting C-2 selective amination of indoles **61** as well as the amination of several heterocycles. Iodobenzene bis(saccharinato) **62** and 8 mol% of molecular iodine were used under black light irradiation (365 nm). The process yielded several C2 aminated indoles **63a**–**e** with a wide scope obtaining chemical yields ranging from poor to excellent (26–98%). A radical process was proposed in this development (Figure 10).

In 2019, Quideau and co-workers [47], described an interesting cyclopropanation reaction using iodine(III)-based dimedone-derived ylides **64** or ylides **65** upon direct irradiation with blue LEDs. The development was systematically conducted starting with (time-dependent) TD-DFT calculations of the HOMO and LUMO to determine the electronic population of the molecule during the time irradiation. Therein, it was identified the biradical nature of iodine(III) and the carbon to which it was bonded. These results indicated that carbene formation by an heterolytic cleavage was not operating.

Then, several reactions of iodonium ylides and different styrenes **66**, gave rise to the formation of dimedone-derived **67** as well as some open 1,3-dicarbonyl-based cyclopropane derivatives **68** in excellent yields. The proposed reaction mechanism started with ylide irradiation to give biradical intermediate **69** or **70**, which, upon reaction with styrenes **66**, generated intermediate **71**, that in a step-way fashion underwent an intramolecularly ring-closing via radical pairing to get the four-membered iodane **72**. Final reductive elimination of iodobenzene yielded observed cyclopropanes **67a**–**f** and **68a** (Figure 11).

In early 2020, Waser and co-workers [48] described a versatile and useful method for the direct *_sp_*^3^C-H alkynylation of aliphatic cyclic ethers and thioethers **73**. The process took place with irradiation of a CFL household bulb using 20 mol% of phenylglyoxalic acid as catalyst and alkynylbenziodoxolones (EBX) reagents **74** as the alkynyl transferring counterpart. The protocol gave rise to the formation of C2-alkynylated cyclic ethers **75**. Overall, the procedure was efficient and with an interesting result when it was applied to cyclic thioethers. In this case, alkynylation on the sulfur atom occurred with the concomitant ring opening, giving the linear fragment **76** with the formation of an alkynyl thioether on one side of the molecule and an aldehyde on the other. Remarkably, no oxidation of the sulfur atom was observed as described by other procedures. Concerning the reaction mechanism, a complex proposal was described. However, the most relevant steps included the formation of acyl radical **77** from phenylglyoxalic acid or its triplet biradical **78** state formation under CFL light. This carried out a HAT at the *_sp_*^3^C2-H of cyclic ethers to form the corresponding alkyl radical **79**. This reacted with EBX derivatives **74**, releasing observed alkynylated products **75a**–**d** along iodanyl radical **79** which continued radical propagation with phenyl phenylglyoxalic acid. On the other side, the same path was proposed for cyclic thioethers, up to radical **79** formation. Therein, an initial reaction with ^3^O_2_ led to the formation of peroxide radical **80** that protonated to get **81** which enabled ring opening to give an aldehyde moiety and sulfur-centered radical **82** that continued its reaction with EBX **74** to yield observed products **76a**,**b** (Figure 12).

In 2023, Chen and co-workers [49] developed a visible-light-induced radical strategy to access 2-oxoindoles containing the fluoromethylensufonyl moiety. Photochemical homolysis of PhI(OCOCH_2_F)_2_ **83** gave rise to **·**CH_2_F radical **85** and 1,4-Diazabicyclo[2.2.2]octane bis(sulfur dioxide) (DABSO) as the SO_2_ source. C3-functionalized oxindoles **84** were synthesized starting from *N*-arylacrylamides **82** in good yields overall. Some relevant features such as catalyst-free, photochemical activation, mild reaction conditions, broad functional group compatibility, and good to excellent yields, highlighted this work. The proposed mechanism started with the generation of fluoromethylene radical **85** by homolysis of a single ligand with iodine(III) reagent **83** giving also iodanyl radical **86**. Formed radical **85** reacted with DABSO to get a new fluoromethylene sulfonyl radical, species **87** which interacted with *N*-arylacrylamide **82** to close the indoline ring, giving rise to non-aromatic radical **89** via intermediate **88**. Final aromatization with iodanyl radical **86** yielded the observed final products **84a**–**e** (Figure 13).

In 2023, Ma and co-workers [50] described the fluorosulfonylation of terminal alkanes using the corresponding carboxylic acids **90** and their carboxyiodobenzene derivatives formed in situ by mixture with PIDA. Photochemical irradiation of these formed reagents with blue LEDs using 4CzIPN (2,4,5,6-tetra-(9*H*)-carbazol-9-yl) as a photocatalyst in combination with DABSO and KHF_2_ as sulfonyl and fluorine sources, respectively, gave rise to the corresponding fluorosulfonylated alkyl derivatives **91**. The observed chemical yields for **91a**–**d** were modest to good while the reaction mechanism was similar to the one proposed by Chen (Figure 14).

A recent report in this section was described by Zhu’s group [51] in 2024. Herein, the direct azolation of inert *_sp_*^3^C-H bonds from cyclic ethers was developed. The rationalization of this process was carried out through a systematic radical polarity analysis framework for the projection of the radical reactivity patterns. Accordingly, blue LED irradiation of PIDA in the presence of benzo[*b*]pyrazoles **92** and different ring-size cyclic ethers **93** led to the formation of observed direct azolation products **94** in good yields. The protocol displayed, in general, good regiochemistry at N1 (Figure 15).

The most recent work at this time was published by Bagdi’s group [52] in 2024. In this study, they described the iodination of imidazopyridines **95** derivatives using PIDA as the iodine(III) reagent, which acted as a source of iodine in this methodology. The system formed by PIDA and *p*-TsOH, under irradiation with blue LED light in the presence of oxygen led to the formation of iodinated products **96a**–**j**. This novel methodology was photocatalyst-free, and its scope included the iodination of electron-donating and electron-withdrawing groups, with iodinated products obtained in moderate to good yields (Figure 16).

Clearly, it can be envisioned that direct photoexcitation applied to bond activation and functionalization has several opportunity areas to expand and other synthetic protocols can be developed. Some identified niches involve the generation of aryl radicals in an easy, controllable, and catalytic way. Also, direct introduction of other functional groups such as formyl or nitro which can undergo the very reactive triplet state through direct photoexcitation are, at this point, excellent opportunities to develop and improve the use of iodine(III) reagents under irradiation in absence of transition metals.

## 4. Photocatalytic Synthetic Methods Involving Transition Metals and Iodine(III) Reagents

According to our outline, the most relevant synthetic protocols involving iodine(III) regents in direct photoredox cycles using transition metals will be reviewed in the next section.

The pursuit of trifluoromethylation of molecules led Zhu’s group [53] in 2013 to develop a protocol for the visible-light-promoted carbotrifluoromethylation of electron-withdrawing alkenes for the preparation of oxindoles bearing a quaternary center. Using Togni’s reagent **98** as the CF_3_ source, *N*-aryl acrylamide derivatives **97** were transformed into their corresponding oxindoles **100** using [Ru(phen)_3_]Cl_2_ (phen=1,10-phenanthroline) **99** as the photocatalyst. The *N*-acrylamide derivatives were protected on the nitrogen atom for the reaction to proceed. The protocol tolerated substituents on the aromatic ring, and even some heterocycles. Also, different α-substituted olefins gave the desired product in moderate to good yields (51–91%). The addition of TEMPO as a radical scavenger inhibited the reaction and just a small amount of product was obtained. This supported the hypothesis that a radical pathway was operative in the process. In the presence of electron-donating groups on the meta position of the aromatic ring, the reaction was regioselective for the most hindered carbon, suggesting that the rate-limiting step was the formation of a cationic intermediate stabilized by the substituents present in the aromatic ring. The proposed mechanism involved the formation of a radical anion **101** from Togni’s reagent via reductive SET enabled by [Ru(bpy)_3_]^2+^. This intermediate collapsed giving rise to formation of a **·**CF3 radical **103** and 2-iodobenzoate **102**. Then, the reaction of **·**CF3 with N-aryl acrylamide derivatives **97** formed carbon-centered radical intermediate **104**. Next, a radical C-H functionalization cascade ensued, forming a radical oxindole **105** which was oxidized through a second SET process catalyzed by Ru(III) to give key cationic intermediate **106**. Final deprotonation of **106**, assisted by **102**, gave rise to the observed products **100a**–**h** (Figure 17).

Chen and co-workers reported in 2014 [54] a photoredox system for the chemoselective *_sp_*^3^C-*_sp_*^2^C coupling of alkyltrifluoroborates **107** and vinyl carboxylic acids **108** by decarboxylative alkenylation using visible light to get the corresponding products **109**. The best results were achieved using the [Ru(bpy)_3_](PF_6_)_2_/acetoxybenziodoxole (BI-OAc) system in DCE/H_2_O. The reaction conditions tolerated alkylboronates with different degrees of substitution and bearing different functional groups such as ketones, esters, and alkyl bromides giving the desired alkenes **109a**–**f** in good yields (62–83%). The vinyl carboxylic acids could bear a wide variety of substituents on the aryl groups of the β-position, producing the final alkene with moderate to good yields (58 to 82%). Reactions with TEMPO confirmed the alkyl radical formation instead of an alkenyl radical. Monitoring the reaction by NMR showed the formation of complex BI-OOCCH=CHR’ **111**, an unprecedented stable species. This was proposed as the first step of the mechanism followed by oxidation of the photoexcited Ru-catalyst which was responsible for the deboronation of the trifluoroalkyl substrate to form alkyl radical **110**. The latter was added to the α-carbon of complex **111** which underwent decarboxylation to generate the benziodoxole radical that gave final alkenes **109a**–**f**. This approach not only had better chemoselectivity compared to the common Heck-type reaction, but the milder reaction conditions opened the possibility to further applications especially for biomolecules (Figure 18).

Indeed, one year later in 2014 [55], Chen’s group expanded their findings to the deboronative alkynylation of alkyl boronates. As its predecessor, the reaction proceeded with a wide variety of alkyl boronates **112**, alkynyl benziodoxole **113**, and hydroxybenziodoxole (BI-OH) in the presence of Na_2_CO_3_ under blue LED (λ_max_ = 468 nm) irradiation using [Ru(bpy)_3_](PF_6_)_2_ as the photocatalyst. The reaction was compatible for alkynes bearing electron-donating and electron-withdrawing groups on the aryl. This was especially relevant for functional groups that were sensitive to other metal-catalyzed reactions, such as aryl halides and azides, among others. The desired alkynes **114a**–**g** were obtained with moderate to good yields (59–82%). Tests with TEMPO were carried out as well as isotopic labeling and on–off light experiments to elucidate the reaction mechanism. The first step of mechanism was the formation of radical **115** enabled by reduction of [Ru(bpy)_3_]^3+^ to [Ru(bpy)_3_]^2+^. The former radical underwent an α-addition to **113** giving rise to β-radical **116** which reacted with BI-OH to produce benziodoxole radical **49** along with the formation of observed products **114**. Radical **49** oxidized *[Ru(II)] to [Ru(III)] allowing it to start a new cycle while 2-iodobenzoic acid **117** was also released. For biomolecule compatibility, it was demonstrated that the reaction could be carried out in pH 7.4 phosphate saline buffer with comparable yields to the DCM/H_2_O system. The presence of aminoacids, nucleosides, oligosaccharides, nucleic acids, proteins, and even bacterial cell lysates did not affect the outcome of the reaction, broadening the application of this protocol to biomolecule research (Figure 19).

Their next report came in 2015 [56] on the decarboxylative ynonylation of alkynes with a combination of hypervalent iodine(III) reagents for the synthesis of ynones, ynamides, and ynoates. The reaction took place submitting α-ketoacids **118** and alkynyl benziodoxole **113** to the [Ru(bpy)_3_](PF_6_)_2_/acetoxybenziodoxole photoredox system in dichloromethane at room temperature for 5 h to give ynones **119a**–**g**. The yields ranged from moderate to excellent, tolerating a wide variety of groups, not only the α-keto acid but also on the alkyne, which was especially interesting for groups such as azides that are usually reactive towards transition-metal-catalyzed reactions. Experiments carried out showed that BI-OAc was not only an oxidant for the photoexcited *[Ru(bpy)_3_]^2+^ species, but it also had a role in the activation of the α-keto acid, forming a benziodoxole/ketoacid complex (BI–O2CCOR’). This protocol could also be modified to run under neutral aqueous reaction conditions, allowing gram-scale synthesis (Figure 20).

Another important development in Chen’s group’s [57] research was the generation of alkoxy radicals by cyclic iodine(III) reagent catalysis reported in 2016. Due to their properties similar to transition-metal reactivity, cyclic iodine(III) reagents were tested as a milder alternative to the harsher conditions necessary when these metal catalysts were used. Specifically, for the formation of alkoxy radicals, it was interesting to promote the *_sp_*^3^C-*_sp_*^3^C cleavage via β-fragmentation. This process, that was usually limited to strained cycloalkanols, could be expanded to even linear alcohols, yielding stable ketones and alkyl radicals that could undergo alkenylation/alkynylation. Thus, alcohols **120** or **121** were transformed into adducts **123** or **124**, respectively, in moderate to good yields (44 to 86%) using acetoylbenziodoxole (BI-OAc) under blue LED (λ_max_ = 468 nm) irradiation and [Ru(bpy)_3_](PF_6_)_2_ as the photocatalyst. Mechanistic studies inferred that the process initiated by oxidation of photoexcited *[Ru(bpy)_3_]^2+^ to [Ru(bpy)_3_]^3+^ with BI-OAc or its decomposing benziodoxole radical **49**. [Ru(bpy)_3_]^3+^ then oxidized benziodoxole/alcohol complex **125** formed in situ and released BI-OAc for a new catalytic cycle. On the other hand, generated alkoxyl radical **126** underwent β-fragmentation to yield alkyl radical **127** which reacted with alkynyl or vinyl carboxylate-bound benziodoxole **128** or **129**. Finally, alkynylated or alkenylated products **123** or **124** were obtained, and benziodoxole radical **49** was released for further *[Ru(bpy)_3_]^2+^ oxidation (Figure 21).

That same year, Li [58] and co-workers published the photoredox-mediated Minisci C-H activation of heteroarenes **130** with alkyl boronic acids **131** using BI-OAc as the oxidant. The reaction proceeded in the presence of 1 mol% of Ru(bpy)_3_Cl_2_. Visible light was used for the photoredox system and the reaction was kept under argon. The reaction showed a broad substrate scope **132a**–**f**, with even primary alkyl boronic acids forming the more challenging primary alkyl radicals **134**. The conditions allowed a great functional group tolerance, including substrates such as alkyl and aryl halides, esters, and carbamates, among others, which opened the possibility for late-stage functionalization for complex molecules. The reaction occurred at C2 or C4 of the heteroarenes with electron-deficient *N*-heteroarenes being more reactive towards alkylation. DFT calculations proposed that after reduction of BI-OAc by photoexcited *[Ru(bpy)_3_]Cl_2_, an *O*-centered radical **133** was formed after I–O bond cleavage. This species reacted with the alkyl boronic acid **131** to form alkyl radical **134** via **133** which underwent the nucleophilic addition reaction with protonated *N*-heteroarenes **136** to form σ-complex **137**. SET oxidation provided final products **138** which were neutralized and closed the photoredox cycle (Figure 22).

Also in 2017 [59], Chen and collaborators described the synthesis of ynamides, ynoates, and ynones from β-amide alcohol **139** and alkynyl benziodoxole **128**. This alkynylation was promoted by the tetrafluoro-BI-OH **140** system and [Ru(bpy)_3_](PF_6_)_2_ as a photocatalyst, which enabled regio- and chemoselective carbonyl-*_sp_*^3^C bond cleavage. Irradiation with blue LED light (468 nm) promoted the *[Ru(II)] to Ru(III) oxidation. This photoredox procedure allowed the photocatalyst to generate radicals in the medium, leading to the key carbonyl-*_sp_*^3^C bond cleavage. The scope of the system enabled the formation of the corresponding products **141a**–**f**, where it was observed that both EDG (electron-donating groups) and EWG (electron-withdrawing groups) were well-tolerated (Figure 23).

Interested in carbyne chemistry, Suero’s group [60] described in 2018 the generation of carbyne equivalents via photoredox catalysis with hypervalent iodine reagents. Benziodoxolone and its pseudocyclic analogue **143** were tested as diazomethyl radical precursors. Mixing this iodine(III) precursor with arene **142** and 1 mol% of [Ru(bpy)_3_](PF_6_)_2_ in CH_3_CN under blue LED irradiation gave the desired C–H diazo methylation. The reaction showed a broad scope, yielding a wide variety of diazomethyl arenes **144a**–**f**. The products were obtained in moderate to good yields, highlighting a successful late-stage functionalization on diverse drugs. Even when the yields were poor, recovery of the original drug was possible while de novo synthesis for those products would be more challenging. Their proposed mechanism involved the formation of diazo methyl radical **146** from hypervalent iodine reagent **143** via SET enabled by oxidation of *[Ru(II)] to [Ru(III)] which also released iodane **145**. Next, this carbyne radical was added to arene **142**, leading to the formation of non-aromatic carbon-centered radical **147** which had the β-diazo ester functionality incorporated. Final oxidative deprotonation of **147** from Ru(III) catalyst gave rise to the formation of observed products **144a**–**f** with concomitant regeneration of [Ru(II)] which entered another catalytic cycle (Figure 24).

Furthermore, Bolm’s group [61] worked on the preparation of new hypervalent iodine(III) reagents to test their reactivity. In 2020, they reported that mixing *p*-tolyldifluoro iodobenzene **149** with sulfoximines **148** and styrenes **151** yielded products **149** under blue LED irradiation and 1 mol% of [Ru(bpy)_3_](PF_6_)_2_ as a photocatalyst. Taking into consideration that both sulfoximine and fluoro groups are of synthetic interest, they tested the reaction of hypervalent-iodine(III) reagent **150** which was formed in situ. A stepwise approach gave the best results when **149** and NH-sulfoximine **148** were stirred in DCM for 20 min under argon at room temperature. Styrene **151** was then added to the reaction mixture and finally irradiated with blue LED (24 W) for 12 h to give the desired product. The reaction showed a broad scope, not only for the sulfoximine where different groups were tolerated on the aryl, but also on the styrene. The reaction showed high regioselectivity for 1,2-disubstituted styrenes due to the benzylic stabilization of radical **153**. Further experiments to elucidate the mechanism were carried out. It was established that the N-I bond cleavage in **150** by single-electron transfer was the first step in the photoredox cycle which generated N-centered radical **153**, iodoaryl **154**, and a fluoride anion. Next, the addition of **151** to the double bond gave rise to C-centered radical **155** that was followed by SET oxidation to form benzylic cation **156**. A final reaction with the fluoride anion yielded observed products **152** (Figure 25).

Inspired by Suero’s work on the generation of diazomethyl radicals, Li’s group [62] described in 2021 a photoredox [3+2] cyclization reaction for the synthesis of 1-amino-1,2,3-triazoles **159** using hypervalent iodine(III) diazo reagents **158**. *N,N*-dialkylhydrazones **157** were transformed into the desired 1,2,3-triazole derivatives under optimized conditions. Hydrazones could bear a wide variety of substituents on the aromatic ring and on the *_sp_*^3^N atom, giving the products in moderate to good yields. The synthetic utility was demonstrated by late-stage functionalization of natural product derivatives. Experimental studies supported a reaction pathway that initiated photochemical excision of diazo-containing iodine(III) reagent **158** enabled by photoexcited *[Ru(bpy)_3_]^2+^ which generated iodoarene **160** and carbon-centered radical **161** containing the diazo moiety. The former radical reacted with hydrazone derivatives **157**, giving homologated nitrogen-centered radical **162** which was oxidized by [Ru(bpy)_3_]^3+^ leading to the formation of the nitrenium **163**. A final attack from the diazo fragment closed the ring, yielding observed final triazoles **159** (Figure 26).

In their follow-up work, Li and co-workers [63] expanded their photoredox-catalyzed [3+2] cyclization using hypervalent iodine(III) reagents to afford [1,2,3]-triazolo-[1,5-a]quinoxalin-4(5*H*)-ones **165**. Their approach was especially attractive since these quinoxalinones are found in many bioactive products but few protocols are available for the synthesis of the tricyclic scaffold with the 1,2,3-triazole moiety. Based on their previous report, hypervalent iodine diazo reagent **158** was used as a diazomethyl radical precursor. Quinoxalinone **164** and 1.5 equivalents of the iodine(III) reagent were mixed with 2 equivalents of sodium acetate and 2.5 mol% of Ru(bpy)_3_Cl_2_ in DCE at room temperature under the irradiation of a blue LED (24 W) for 12 h. The conditions tolerated a wide variety of N-substituents on the quinoxalinone core, giving tricyclic compounds **165** with moderate to good yields. Limitations of the process imply the free nitrogen did not afford the desired product. The reaction proceeded with different substituents of the aromatic ring except when an NO_2_ group was present at the C6-position. Different ester and benzoyl groups were introduced on the hypervalent iodine(III) diazoderivative **158** reagents showing good performance. Further experiments to elucidate the mechanism supported the formation of the diazoacetate radical which could attack the C=N bond of the quinoxalinone, forming another radical that, upon oxidation with [Ru(bpy)_3_]^3+^, would form the cation that would provide the tricyclic product in the presence of a base. Overall, it was quite similar to the mechanism reported on their previous work (Figure 27).

In 2022, Chen’s group [64] proposed that BI-OAc could generate amidyl radicals via N-N bond cleavage of amidyl-iminophenyl-acetic acids (IPA-NR_2_) that in a Minisci-type reaction, promoted the *_sp_*^2^C-H amination of arenes and heteroarenes. IPA-NR_2_ **166** were easily accessible through condensation of hidrazines with methyl benzoylformates. The reaction between them and heteroarenes **167** proceeded in the presence of 2 equivalents of BI-OAc, 2 mol% of [Ru(bpy)_3_](PF_6_)_2_ in acetonitrile under blue LED irradiation for 12 h to give 2-aminoindoles **168**. In this way, products were obtained with moderate to good yields. Electron-deficient and electron-rich arenes and heteroarenes performed in a similar way under the reaction conditions and N-substitution even favored it. On the other hand, simple benzenes, pyridines, or isoquinolines were not suitable and gave hydrogenation adduct TsNHMe as the product. Different sulfonamide substituents on the IPA were well-tolerated. The importance of BI-OAc coordination to IPA-NR_2_ was demonstrated by carrying out the reaction with the corresponding methyl ester which did not react. The mechanistic studies were consistent with previous reports about cyclic iodine activation of carboxylates. Thus, initial interaction between **166** and BI-OAc formed iodine(III) intermediate BI-IPA-NTsMe **169** which was oxidized to the corresponding iminophenylcarboxyl radical cation **170** by Ru(III) photocatalyst, then underwent concerted carbon dioxide and phenylnitrile elimination to form sulfonamidyl radical **171**. Radical addition to the arene **167**, followed by oxidation with BI radical afforded final products **168a**–**d** (Figure 28).

Most recently, in 2024, Suero’s group [65] reported the first photoredox-catalyzed alkoxy diazomethylation of alkenes, yielding compounds from the elusive β-alkoxydiazo family. Following a similar protocol previously described by his group, the use of alkenes **172** as substrates could allow the introduction of the ether moiety by trapping the final cation with an alcohol in a multicomponent disconnection approach. Indeed, the reaction proceeded using diazo-containing hypervalent iodine(III) reagent **173** and [Ru(dtbbpy)_3_](PF_6_)_2_ (dtbbpy = 4,4′-di-*t*-butyl-2,2′-bipyridine) which had a better reductive capability than [Ru(bpy)_3_](PF_6_)_2_ in its excited state, hence increasing the efficiency of the radical generation to get adducts **174**. The reaction tolerated a wide variety of styrenes with different functionalities in every position of the aromatic ring with moderate to excellent yields; even α-alkyl substituted styrenes worked well in the conditions of the reaction. The reaction with α,β-di-substituted styrenes gave E/Z mixtures with high yields and excellent diastereoconvergence. Alcohols with different degrees of substitution were introduced with high yields and even an intramolecular version was achieved. Finally, the evaluation of the scope on the iodine reagent showed that derivatives with benzyl esters and even phosphonate and sulfonate were suitable in these conditions. The reaction mechanism was similar to the one previously described (Figure 29).

The most recent example using iodine(III) reagents in a photocatalytic cycle using iridium, came from Zhu and co-workers [66] in 2013. Herein, *N*-methyl-*N*-phenylmethacrylamide **175** and different alkyl carboxylic acids **176** were used for a sequential decarboxylative/C-H functionalization reaction that produced C3-disubstituded 2-oxoindoles **177**. The scope covered the functionalization of acrylamides with different alkyls. The proposal of the reaction mechanism started with the *[Ir^III^(ppy)_3_]-promoted (ppy = 2-phenylpyridine) photochemical reduction of the iodine(III)-containing alkyl group to form iodanyl radical **178**. The following reaction of this iodine-centered radical with acrylamide gave rise to adduct **179** which underwent homolytical cleavage and transferred the alkyl fragment to the double bond of the amide, leading to carbon dioxide and iodobenzene release, along to the formation of carbon-centered non-aromatic 2-oxoindole radical **180**. This intermediate was oxidized by [Ir^IV^(ppy)_3_]^+^ to its corresponding carbocation **181**. The final acetate assisted aromatization and yielded observed products **177a**–**i** (Figure 30).

## 5. Summary and Perspectives

In summary, several synthetic protocols have been developed, mainly the last 20 years using this powerful combination of iodine(III) reagents and photochemistry. The energy needed to irradiate the reaction mixtures has been easily controlled by setting up the wavelength. Plenty of valuable characteristics can be listed such as very mild reaction conditions, easy to handle reagents, and, remarkably, almost all of the protocols are carried out at room temperature (23–25 °C) or 0 °C, but never require heating conditions. Additionally, several of the iodine(III) intermediates or reagents have been isolated due to their stability, which simplifies the analysis of all the reaction pathways, allowing even to complete mechanistic studies with relative ease.

Through the iodine(III)-photochemistry combination, several functional groups have been introduced in a radical way such as trifluoromethyl, alkyl, alkenyl, alkynyl, fluoromethylene, or fluorosulfonyl to name a few, in different cores including aromatics, alkenyls, or alkyls. This strategy resulted unique in terms of innovation and easiness to directly break and make high-in-energy bonds, which would result at worst unviable, or require several steps using other protocols.

Due to the effectiveness of this synergic combination, an interesting perspective which is particularly relevant, involves the aryl–aryl and aryl–heteroaryl cross-bond formation. This plausible development would have a strong impact on organic synthesis, since essentially it would represent the full emulation of common transition-metal chemistry, providing a full and comprehensive tool that could be tuned deliberately according to synthetic necessities.

## Data Availability

The data presented in this study are available on request from the author.

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
