# Peer review of "Iodine(III)-Containing Reagents in Photo-Assisted and Photo-Catalyzed Organic Synthesis"

_molecules, 2025, doi:10.3390/molecules30040784_

Round 1
Reviewer 1 Report
Comments and Suggestions for Authors
1. The review article primarily focuses on presenting existing research findings without adequately highlighting the unresolved issues and potential areas for further exploration. To enhance the article's value, it is recommended that the author explicitly identify these gaps in the literature and propose future research directions.
2. Throughout the paper, the author provides objective commentary and conclusions but fails to offer personal insights or expectations for the future development of the field. Incorporating the author's perspective and predictions could add depth and uniqueness to the review.
3. While the article covers a broad range of topics, it lacks coherence between the various issues discussed. The context and connections between these points are not clearly established, resulting in a somewhat disjointed narrative. To improve readability and comprehension, the author should strive to create a more cohesive flow between the different sections.
4. The labeling in Figure 1 is not sufficiently clear, which may hinder readers' understanding of the presented data or information. It is recommended that the author revise the figure to improve the clarity of the labels and ensure that they accurately represent the intended content.
Comments on the Quality of English Language
There are no conflicts of interest.
Author Response
Reviewer 1:
- The review article primarily focuses on presenting existing research findings without adequately highlighting the unresolved issues and potential areas for further exploration. To enhance the article's value, it is recommended that the author explicitly identify these gaps in the literature and propose future research directions.
Observation is right, thanks. Accordingly, it was added at the end of the “direct photoexcitation” section, some plausible opportunity areas that can mark future directions of this synergic chemistry.
- Throughout the paper, the author provides objective commentary and conclusions but fails to offer personal insights or expectations for the future development of the field. Incorporating the author's perspective and predictions could add depth and uniqueness to the review.
In the section 5, Summary and perspectives, I comment this point that the referee kindly suggest, adding by far, a couple of words for indicating is a personal perspective.
- While the article covers a broad range of topics, it lacks coherence between the various issues discussed. The context and connections between these points are not clearly established, resulting in a somewhat disjointed narrative. To improve readability and comprehension, the author should strive to create a more cohesive flow between the different sections.
We added some text in every section attempting to address the referee suggestion.
- The labeling in Figure 1 is not sufficiently clear, which may hinder readers' understanding of the presented data or information. It is recommended that the author revise the figure to improve the clarity of the labels and ensure that they accurately represent the intended content.
The figure 1 was already commented in the footnote to clarify the information.
Reviewer 2 Report
Comments and Suggestions for Authors
First of all I would like to underline that the topic is important and the review paper should be given a chance to be published in Molecules. The current manuscript however requires a major revision before being reconsidered for publication. These are my critical points:
1. The title 'Iodine(III) in photochemistry' is too general (based on the true content of the review) and slightly misleading. I would propose an alternative: Iodine(III)-containing reagents in photo-assisted and photocatalyzed (oxidative) organic synthesis'. Also, a brief overview does not correspond with 26 A4 pages of text.
2. This is a review paper and I am therefore concerned about the 14 names as the co-authors. I am not sure about the editorial rules in Molecules, but RSC or ACS journals request a list of detailed justification of individual significant contributions. For research papers perhaps, but for review papers it is highly unusual to have such a big team summarizing a rather limited realm.
3. Introduction (in a broader context) and, in particular, Summary and Perspectives, are too concise and vague at places.
4. English syntax and orthography need to be revised substantially (by a professional native English speaker) - see my comments below.
5. I have a number of a minor, but still important points and suggestions that follow:
a/ p.1, Abstract
- 'green oxidant' is a slang term - it should be explained properly
- photoredox cycles using some transition metals - in fact, not bare metals but their coordination compounds (polypyridyl complexes), with Ru(II) and Ir(III) centres undergoing sensitized photooxidation to Ru(III) and Ir(IV).
b/ - p.2, Eq. (1) - energy should be expressed in wavenumbers and cm-1;
- p.2 - Figure 1 is diffuse, and in fact redundant. Literature reference should be given in the captions to those who did the calculations. Has the picture been reproduced?
c/ p.3 - zwitterionic iodonium salts 7 and 1.3-oxathiole-2-thione 8 - the numbering does not correspond with Eq. (2).
- ibutyl (vinyl ether): i-butyl or iso-butyl
d/ p.4 - Scheme 2 - piridine's core functionalization = pyridine-core functionalization
- Scheme 3 - Mg - hv = Hg
e/ p. 7 - explain BI-OH also in the main text; Schemes and Figures need to be independent of the main text
f/ In general, citing literature, use 'name-XY et al.' (first author) or 'name-XY and co-workers' (research lead), or 'the group of name-XY', but refrain from using 'name-XY' only, as this regards only a single author.
g/ p.8, l.197 - blue led = blue LED; also on p.15 and p.16 468+-25 nm - use only 468 nm for lambda(max).
g/ Starting on p.14, there are many mistakes in writing the formulas of coordination compounds (using square brackets is mandatory):
correct formulas:
[Ru(phen)3]Cl2, [Ru(bpy)3]2+, [Ru(bpy)3](PF6)2;
excited states - *[Ru(bpy)3]2+, *[Ru(bpy)3]Cl2, *[Ir(ppy)]
mistakes:
using byp instead of bpy; phen = 1,10-phenathroline; Ru(byp)32+; Ru(bpy)3Cl2, Ru(bpy)33+; [Ru(bpy)32+]*; [Ru(bpy)3Cl2]*, [Ru(bpy)3](PF6) (Scheme 18), Ru(bpy)2(PF6)2 (Scheme 28), etc.
h/ p.24, Scheme 29 - dtbbpy not defined (4,4'-di-t-butyl-2,2'-bipyridine)
i/ Use full formulas *[IrIII(ppy)3] and [IrIV(ppy)3]+ instead of undefined [IrIII]* and [IrIV].
j/ DABSO defined in Scheme 14, but it also occurs in preceding Scheme 13.
k/ Schemes 14 and 15 - define PIDA (independently of the main text).
As recommended, the revised manuscript should be controlled and approved by a native English-speaking (professional) colleague. Even though the text reads well in general, the syntax can be improved at many places to avoid confusion. The orthography is another critical point - there have been many typos.
For example, only in Summary (p.26):
poweful
fluoromethylen
high enegy
efectiviness
corrs-bond
Author Response
Reviewer 2:
First of all, I would like to underline that the topic is important and the review paper should be given a chance to be published in Molecules. The current manuscript however requires a major revision before being reconsidered for publication. These are my critical points:
- The title 'Iodine(III) in photochemistry' is too general (based on the true content of the review) and slightly misleading. I would propose an alternative: Iodine(III)-containing reagents in photo-assisted and photocatalyzed (oxidative) organic synthesis'. Also, a brief overview does not correspond with 26 A4 pages of text.
We thank to the reviewer comment. We agree with the proposed title change, also regarding it is not a brief overview, we remove these words of the title
- This is a review paper and I am therefore concerned about the 14 names as the co-authors. I am not sure about the editorial rules in Molecules, but RSC or ACS journals request a list of detailed justification of individual significant contributions. For research papers perhaps, but for review papers it is highly unusual to have such a big team summarizing a rather limited realm.
We kindly appreciate the observation, and we take the comment as such, with all the scientific respect to provide an advice for improving the quality to this submitted manuscript. Nevertheless, I couldn´t find any restrictive rule about ““number” of names in a “review” manuscript.
The concern is to me in the sense, why my academic peers are questioning if my criteria on involving authors in a manuscript is the correct one?
Academic formation of scientists under my guidance includes this class of experience about looking for information, illustrate it as well as writing a discussion which will be supervised.
If editors consider in a justified way I need to remove names, in principle I will analyze the proposal, then and only in such case I will consider. Meanwhile, attached to this document is an author list with a detailed contribution activity.
- Introduction (in a broader context) and, in particular, Summary and Perspectives, are too concise and vague at places.
We complete the manuscript in the requested sections, providing from our perspective, the most relevant and critical points to be considered.
- English syntax and orthography need to be revised substantially (by a professional native English speaker) - see my comments below.
All the manuscript was carefully reviewed by a native English speaker involved in organic chemistry area. Mistakes found were properly corrected and highlighted in red color.
- I have a number of a minor, but still important points and suggestions that follow:
a/ p.1, Abstract
- 'green oxidant' is a slang term - it should be explained properly
Explanation for this concept was included in brackets immediately after the word.
- photoredox cycles using some transition metals - in fact, not bare metals but their coordination compounds (polypyridyl complexes), with Ru(II) and Ir(III) centres undergoing sensitized photooxidation to Ru(III) and Ir(IV).
- The scheme have been verified and we have followed the instructions.
b/ - p.2, Eq. (1) - energy should be expressed in wavenumbers and cm-1;
the properly equation was included in the text of manuscript
- p.2 - Figure 1 is diffuse, and in fact redundant. Literature reference should be given in the captions to those who did the calculations. Has the picture been reproduced?
We appreciate your comments, the image has been replaced by a better one. Figure 1 is a supplement for readers, this figure is our authorship interpreting the calculations found in the following review: 27. Xue, X.-S.; Ji, P.; Zhou, B.; Cheng, J. P. The Essential Role of Bond Energetics in C–H Activation/Functionalization. Chem. Rev. 2017, 117, 8622-8648. https://pubs.acs.org/doi/10.1021/acs.chemrev.6b00664
This reference is correctly cited in that paragraph, in the manuscript that you have reviewed.
c/ p.3 - zwitterionic iodonium salts 7 and 1.3-oxathiole-2-thione 8 - the numbering does not correspond with Eq. (2).
We appreciate your comments, the numbering has been corrected.
- ibutyl (vinyl ether): i-butyl or iso-butyl
We addressed the comment and corrected it: i-butyl (vinyl ether). Thank you for the comment, we made the suggested modification.
d/ p.4 - Scheme 2 - piridine's core functionalization = pyridine-core functionalization
Thank you for the comment, we made the suggested modification.
- Scheme 3 - Mg - hv = Hg
Thanks for your comment, scheme 3 has been corrected.
e/ p. 7 - explain BI-OH also in the main text; Schemes and Figures need to be independent of the main text
Thank you for your comment, we added the description of the compound and represented it like this in the text: hydroxybenziodoxolone (BI-OH).
f/ In general, citing literature, use 'name-XY et al.' (first author) or 'name-XY and co-workers' (research lead), or 'the group of name-XY', but refrain from using 'name-XY' only, as this regards only a single author.
Thank you for your comment, we made the suggested modifications for the authors we discussed in the text.
g/ p.8, l.197 - blue led = blue LED; also on p.15 and p.16 468+-25 nm - use only 468 nm for lambda(max).
We attended to the corrections on pages 9, 15 and 16.
g/ Starting on p.14, there are many mistakes in writing the formulas of coordination compounds (using square brackets is mandatory):
correct formulas: [Ru(phen)3]Cl2, [Ru(bpy)3]2+, [Ru(bpy)3](PF6)2;
excited states - *[Ru(bpy)3]2+, *[Ru(bpy)3]Cl2, *[Ir(ppy)]
mistakes: using byp instead of bpy; phen = 1,10-phenathroline; Ru(byp)32+; Ru(bpy)3Cl2, Ru(bpy)33+; [Ru(bpy)32+]*; [Ru(bpy)3Cl2]*, [Ru(bpy)3](PF6) (Scheme 18), Ru(bpy)2(PF6)2 (Scheme 28), etc.
We have adjusted each of the points raised in this comment, thank you for these observations.
h/ p.24, Scheme 29 - dtbbpy not defined (4,4'-di-t-butyl-2,2'-bipyridine)
We have abandoned this comment, in the text we have included the description requested here as well as that of (bpy = 2,2'-bipyridine)
i/ Use full formulas *[IrIII(ppy)3] and [IrIV(ppy)3]+ instead of undefined [IrIII]* and [IrIV].
We have made the changes suggested in the comment, in turn we added the (ppy = 2-phenylpyridine).
j/ DABSO defined in Scheme 14, but it also occurs in preceding Scheme 13.
Thanks for the comment, the schemes were adjusted and the text was added 1,4-Diazabicyclo[2.2.2]octane bis(sulfur dioxide) (DABSO).
k/ Schemes 14 and 15 - define PIDA (independently of the main text).
Thank you for your comments, the relevant changes have been made.
Reviewer 3 Report
Comments and Suggestions for Authors
This paper summarized the latest applications of iodine reagents in photochemical conversion, covering direct irradiation and photocatalytic redox cycles involving transition metals, and looked forward to future development directions. It is acceptable after minor revision.
1.There are numerous expression errors in the article. For instance, in the sentence ‘The following procedure was described by Varvoglis who described the UV - photoexcitation of zwitterionic iodonium salts 7 which gave 1,3 - oxathiole - 2 - thione 8 in reaction with carbon disulfide’, the digit ‘7’ therein should be rewritten as ‘8’, and correspondingly, ‘8’ should be revised to ‘9’.
In Scheme 19, ‘94e’ has been erroneously recorded as ‘94a’. In Scheme 28, the raw material ‘166’ was incorrectly stated as ‘168’.
In the sentence ‘The developed procedure took place via radical decarboxylation of the adamantyl - containing iodine(III) reagent 17 to give a carbon - centered radical 20 which reacted with the disulfide 18 to finally give the formation of sulfides 19a - f’, ‘19a - f’ ought to be changed to ‘19a - d’. In the sentence ‘This intermediate is oxidized by [IrIV] catalyst to its corresponding carbocation. Final acetate assisted aromatization yielded the observed products 177a - k (Scheme 30)’, ‘177a - k’ should be rewritten as ‘177a - i’.
2.There are also multiple formatting problems in the article. For example, in the sentence ‘In 2009, Matveeva carried out the synthesis of several λ5 - phposhinolines 34 and 35 by the photochemical reaction between acetylenes 33 and mixed phosphonium - iodonium ylides 31 or 32’, the font of ‘33’ should be bolded. The same issue exists with ‘60’, ‘a - f’, and ‘76 a - b’. In the sentence ‘In 2023, Ma and Liu, described the fluorosulfonylation of terminal alkanes using the corresponding carboxylic acids 90 and their carboxyiodobenzene derivatives formed in situ by mixing with PIDA’, the font of ‘90’ is incorrect.
Author Response
Reviewer 3:
This paper summarized the latest applications of iodine reagents in photochemical conversion, covering direct irradiation and photocatalytic redox cycles involving transition metals, and looked forward to future development directions. It is acceptable after minor revision.
1.There are numerous expression errors in the article. For instance, in the sentence ‘The following procedure was described by Varvoglis who described the UV - photoexcitation of zwitterionic iodonium salts 7 which gave 1,3 - oxathiole - 2 - thione 8 in reaction with carbon disulfide’, the digit ‘7’ therein should be rewritten as ‘8’, and correspondingly, ‘8’ should be revised to ‘9’.
The numbers corresponding to the text already with the scheme
In Scheme 19, ‘94e’ has been erroneously recorded as ‘94a’. In Scheme 28, the raw material ‘166’ was incorrectly stated as ‘168’.
The schematic has been edited and the numbers corresponding to the text have been corrected
In the sentence ‘The developed procedure took place via radical decarboxylation of the adamantyl - containing iodine(III) reagent 17 to give a carbon - centered radical 20 which reacted with the disulfide 18 to finally give the formation of sulfides 19a - f’, ‘19a - f’ ought to b changed to ‘19a - d’. In the sentence ‘This intermediate is oxidized by [IrIV] catalyst to its corresponding carbocation. Final acetate assisted aromatization yielded the observed products 177a - k (Scheme 30)’, ‘177a - k’ should be rewritten as ‘177a - i’.
The numbers corresponding to the text have already changed
2.There are also multiple formatting problems in the article. For example, in the sentence ‘In 2009, Matveeva carried out the synthesis of several λ5 - phposhinolines 34 and 35 by the photochemical reaction between acetylenes 33 and mixed phosphonium - iodonium ylides 31 or 32’, the font of ‘33’ should be bolded. The same issue exists with ‘60’, ‘a - f’, and ‘76 a - b’. In the sentence ‘In 2023, Ma and Liu, described the fluorosulfonylation of terminal alkanes using the corresponding carboxylic acids 90 and their carboxyiodobenzene derivatives formed in situ by mixing with PIDA’, the font of ‘90’ is incorrect.
The numbers corresponding to the text have already changed
Round 2
Reviewer 1 Report
Comments and Suggestions for Authors
All of problems have been addressed.
Reviewer 2 Report
Comments and Suggestions for Authors
This revised v.2 of the manuscript has addressed all critical points of the reviewer appreciably. The only remaining minor amendment regards Equation (1).
In the original manuscript:
E= 2.86 x 104 / λ (in cm-1)
In the revised manuscript:
E(cm-1)= 349.8 cm-1 /kcal/mol
The wavenumber (in cm-1) is represented by the Greek letter nu (v) with a tilde (~) over it. The wavelength is resented by λ (in nm).
I proposed to replace in original Equation (1) 1/λ with (v~).
Alternatively, E (cm-1) = 0.002859 E (kcal mol-1).
Equation (1) in the revised manuscript does not give sense to me.
Perhaps keep the original version:
E = h v~ = 2.86 x 104 / λ (in cm-1)